# Effect of Soil Type on Native *Pterocypsela laciniata* Performance under Single Invasion and Co-Invasion

**DOI:** 10.3390/life12111898

**Published:** 2022-11-15

**Authors:** Rakhwe Kama, Qaiser Javed, Yuan Liu, Zhongyang Li, Babar Iqbal, Sekouna Diatta, Jianfan Sun

**Affiliations:** 1Institute of Environment and Ecology, School of the Environment and Safety Engineering, Jiangsu University, Zhenjiang 212013, China; 2Institute of Farmland Irrigation of CAAS, Xinxiang 453002, China; 3National Research and Observation Station of Shangqiu Agro-Ecology System, Shangqiu 476000, China; 4Laboratory of Ecology, Faculty of Sciences and Technology, Cheikh Anta Diop University of Dakar, Dakar 50005, Senegal

**Keywords:** soil diversity, single invasion, co-invasion, *Pterocypsela laciniata*, soil type

## Abstract

Native and invasive plant competition is usually controlled by the soil properties and the soil type underlying interspecific interactions. However, many uncertainties exist regarding the impact of soil type on native plant growth under single invasion and co-invasion despite the significant number of previous studies that focused on plant invasion. This study aims to elucidate the effects of soil type on the response of the native plant *Pterocypsela laciniata* under single invasion and co-invasion. Three different soils were used: natural soil, nutrient soil, and nutrient sterilized soil. The native *P. laciniata* was grown in monoculture and under single invasion and co-invasion with *Solidago canadensis* and *Aster subulatus* Michx. The results show that the native plant height and total biomass were 75% and 93.33% higher, respectively, in nutrient sterilized soil in monoculture than in natural and nutrient soil. In contrast, the native *P. laciniata* presents its best competitive ability in nutrient sterilized soil, being about 100% higher than in natural and nutrient soil under single invasion and co-invasion. However, no significant increase was observed in its growth parameters under co-invasion compared to single invasion. Conclusively, this study shows that nutrient soil sterilization positively affects native plant growth in monoculture and under single invasion, contrasting co-invasion in which more pronounced negative effects were observed on the native plant response.

## 1. Introduction

Exotic plant invasion is a crucial part of ecosystem structure and dynamics [1], underlying ecosystem processes and functions [2]. The impacts of plant invasion on ecosystem biodiversity have expressively raised the awareness of researchers and the general public over the last three decades [3]. In addition, several studies have focused on invasive plant control and management and their effects on native communities [4,5]. It has been revealed that the invasion of alien/exotic plant species causes severe ecological threats to native biodiversities, such as altering ecosystem services, affecting human and animal health, and interfering with agricultural production with a significant decrease in arable land productivity [6,7]. The success of invasive plants in their new habitats is mainly due to their rapid dispersal or establishment [8,9]. Several studies on invasive plant control and ecosystem management focused on invasive plants neglecting the native plant response. Many uncertainties still exist regarding the native plant response to these invasive plants’ threat under different soil conditions.

Co-invasion is a general and popular phenomenon in natural ecosystems, but it has been mostly neglected in many case studies regarding competition and plant invasion. Co-invasion results in character displacement or the character convergence of invasive plants [10]. For instance, large population sizes and the genetic diversity of native species allow them to evolve traits in order to co-occur with invasive species [11]. Under some circumstances, such as limited resource availability, species are expected to evolve and use the same resources, leading to a convergence instead of a divergence of traits, which promotes the co-evolution of invasive plants with native plants [12]. Invasive species can trigger prominent changes in the arrangement and function of native ecosystems [13] and consequently disturb native plant growth. Previous studies have suggested that the native plant response under invasion depends on the plant itself, the invader’s identity, and the soil conditions. In addition, most studies on ecosystem diversity and invasive plant effects on native plant growth focused on single invasion and/or soil types [14,15]. However, the native plant response under co-invasion in different soils is less understood.

Plant interaction is an important and complex phenomenon underlying communities, and it is usually mediated by soil composition and nutrient availability [16]. However, the effects of soil type and properties, including microbial structure and diversity, on the native plant response under invasion are still less understood despite the evidence of a strong linkage between the aboveground and belowground biota. Several studies have suggested that the soil biota in some invaded ecosystems may facilitate exotic plant invasion [17,18,19,20] or reduce the potential novel biochemical interactions between native and invasive plants. However, little is known about plant–plant interactions and the soil type’s impact on the native plant response under single invasion and co-invasion. This study aims to evaluate the effect of invasive plant presence and soil type on the response of the native *Pterocypsela laciniata* under single invasion and co-invasion.

Several studies have focused on the success of invasive plants over native plants. For instance, one finding provided evidence that soil invaded by *Solidago canadensis* suppresses its growth and competitive ability against native neighbors in the feedback [21]. Another finding indicated that plant–soil interactions shift the competitive results between native and invasive species or benefit the invasive species [22].

Studies have been carried out on invasive plant management control by considering the response of invasive and native plants but without considering the impact of soil diversity. To elucidate and understand invasive plant control, it is also necessary to develop new mechanisms in order to support the effectiveness of the previous methods. We believe that understanding the native plant response under single invasion and co-invasion in different soil conditions could be an opportunity to improve the understanding of the behavior of invasive species. Therefore, to fil the gap in this research area, it is necessary to assess the effects of soil type on the native plant response based on the invasive plant identity and the type of invasion.

## 2. Materials and Methods

### 2.1. Study Species

All three plant species, viz., *Pterocypsela laciniata*, *Solidago canadensis* L., and *Aster subulatus* Michx, used for the study are perennials with similar morphological and biological characteristics, but they have different origins [23,24]. *P. laciniata* is native to China and is mainly distributed in Shandong, Zhejiang, and Jiangxi Provinces [24]. *S. canadensis* presents a strong expansion range, and it is considered a serious invasive weed widely distributed in the eastern US and Canada [23]. It negatively affects native ecosystems in many countries, especially in Europe and Asia [25]. *Aster subulatus* Michx is native to North America but invasive in China, and it is widely distributed in Jiangsu, Hunan, Hubei, and Sichuan Provinces [26]. Both *S. canadensis* and *A. subulatus* Michx have become invasive worldwide, causing serious environmental problems in many countries.

### 2.2. Soil Preparation and Seed Collection

Three different soils (natural soil, nutrient soil, and nutrient sterilized soil) and three species were used in this study. The natural soil was sandy loam soil (specifications: pH 6.0, organic matter of 1.65%, total nutrients of 1.5%, water content of 20%, and electrical conductivity of 1.75 ds/m), and it was collected from a natural field near Jiao Shan Mountain in Zhenjiang city, Jiangsu Province; the nutrient soil (specifications: pH 7.0, organic matter of 40%, total nutrients of 4.0%, water content of 25%, and electrical conductivity of 2.5 ds/m), characterized by a high level of macro- and micro-nutrients, was purchased online, and the nutrient soil’s sterilization was carried out using autoclave, with the pressure maintained at 15 Lbs and the temperature maintained at 120 °C for 30 min. The seeds of *P. laciniata* were collected from adult plants in Zhenjiang, Jiangsu Province, near the roadside in October 2018. The seeds of *S. canadensis* and *A. subulatus* Michx were collected near Jiaoshan Mountain in Zhenjiang city, Jiangsu Province, in November 2018. All seeds were stored in paper bags until March 2019.

### 2.3. Experimental Design

A complete, randomized design was used to explore the native *P. laciniata* response under single invasion and co-invasion in three soil types. The native plant was grown in monoculture and under single invasion, and it was grown under co-invasion with the two invasives *S. canadensis* and *A. subulatus* Michx in natural soil, nutrient soil, and nutrient sterilized soil; each treatment was replicated five times. In this design, pot numbers were used as the random factors, and the growth parameters were the dependent factors. A pot experiment was set up using different planting patterns and different soil types in a greenhouse from May to July 2019. Plant seedlings were planted separately in seedbeds, and consistently growing seedlings were selected and transplanted into pots (filled with 2 kg of each type of soil) after two weeks of growth. The number of native plant seedlings per pot was two in the monoculture and one under single invasion and co-invasion. Watering was carried out every two or three days with tap water to maintain the soil moisture at 60–70% of the water holding capacity.

### 2.4. Plant and Soil Data Measurements

The soil pH, electrical conductivity, moisture content, plant height, and stem diameter were recorded every three days, 20 days after transplanting. The plants were harvested after two months, and each replicate’s shoot and root biomass were collected. Mean values were obtained per plant and used for statistical analyses. The considered and measured data for the statistical analyses were plant height, stem diameter, soil electrical conductivity (EC), soil temperature, pH, and moisture. Plant height was measured with a tape and was recorded as the distance from the ground to the highest leaf position. The stem diameter was measured using a Vernier caliper. The soil EC, moisture, and temperature were measured with a moisture meter (TR-6/TR-6D), while pH was measured using a pH meter (AMT-300). The shoot and root biomass were obtained with the weight balance. All the data were recorded by directly inserting the instruments into the soil in the pots.

### 2.5. Total Phenolic Content

The extract’s total phenolic content (mg/g) was determined using the Folin–Ciocalteu method [27]. Briefly, 2 g of soil was weighed, placed in a test tube, ultrasonically extracted with 10 mL of 70% ethanol for 30 min, heated in a water bath at 70 °C for 30 min, filtered, and added to the ethanol solution (the solution was made up of 10 mL of 70% ethanol and 1 mL of the extract, which was diluted with 10% ethanol to a volume of 10 mL. Then, 1 mL of 0.2 M Folin reagent was added, shook, and left to stand for 2 min, and finally, 1 mL of 2% Na_2_CO_3_ was added). The mixture was allowed to stand for 60 min in the dark, and absorbance was measured at 650 nm. The total phenolic content was calculated from a calibration curve, and the results are expressed as mg of Gallic acid equivalent per g dry weight (mg/g).

### 2.6. Total Flavonoid Content

Briefly, 2 g of the soil sample was weighed, placed in a 10 mL stoppered test tube, soaked in 6 mL of 70% ethanol for 24 h, placed in a 70 °C water bath, and heated to reflux. After 60 min, the flavonoid extract was obtained by filtration to a constant volume. The mixture was allowed to stand for 15 min, and, then, 0.5 mL of 1 mol/L NaOH was added, with the total being made up to 2.5 mL with distilled water. The solution was mixed well, and the absorbance was measured at 510 nm. Afterwards, 1.0 mL of the total flavonoid was pipetted into a 10 mL volumetric flask and added to the subsequent solutions that were made of ethanol, NaNO_2_, Al (NO_3_)_3_, and NaOH to the standard. The total flavonoid content was calculated from a calibration curve, and the results are expressed as mg rutin equivalent per g dry weight (mg/g) [28].

### 2.7. Root/Shoot Ratio

The root/shoot biomass ratios were determined according to [29] by using the following equation, where *R* is the dry weight of the root (g), and *S* is the dry weight of the shoot (g):(1)Root−shoot biomass ratio=RB/SB

### 2.8. Competitive Interaction Index

The competitive interaction index (CII) of the native target plant in monoculture and mixed culture was calculated based on the total dry weight of the plant. The CII is appropriate for evaluating the positive and negative interactions among two species. By using the CII, the performance of *P. laciniata* under different plant patterns depending on the invasive plant identity and diversity, as well as their combinations, can be compared. The CII was calculated [30,31] using the following equation:(2)CII=(Ax1x2−Ax1)/(Ax1x2+Ax1)
where *A* is the total dry weight of the native plant *P. laciniata*, *x*1 represents the total dry weight of the native plant in the control (CK, no competition with other invasive plants), and *x*2 represents the total dry weight of the native plant in competition with the invasive plant.

### 2.9. Statistical Analysis

SPSS version 22:0 (SPSS Inc., Chicago, IL, USA) was used for statistical analyses, and Origin 2019b was used for figure production. The differences in native plant height and root and shoot biomass grown in monoculture and under single invasion and co-invasion were tested using an analysis of variance (ANOVA) to determine the changes in the native plant parameters, followed by multiple comparisons tests, i.e., Duncan’s multiple-range test and Tukey’s test. Data are presented as means ± SE at *p* < 0.05. All the treatments were grouped for the mean separation, and differences were compared among plant combinations, i.e., monoculture against mixed culture in the three types of soils. All the data are based on the mean values.

## 3. Results

### 3.1. Effects of Invasive Plant Identity on Native P. laciniata Growth under Single Invasion and Co-Invasion

Significant differences were observed in the native plant height and stem diameter under invasion compared with monoculture (Figure 1). A better growth rate and competitive ability were observed in the nutrient sterilized soil, followed by the natural soil. The results show that the native plant height was 75% higher in the nutrient sterilized soil in monoculture than in the nutrient soil. At the same time, the stem diameter of *P. laciniata* was noted to be 42.85% higher in the natural and nutrient sterilized soils in monoculture than that in the natural and nutrient sterilized soils under co-invasion.

In addition, more pronounced negative effects of the invasive plants on the native *P. laciniata* were observed under *A. subulatus* Michx invasion than under *S. canadensis* invasion. This negative effect of invasive plant presence was more significant under co-invasion than under single invasion.

### 3.2. Effects of Soil Diversity on Native Plant Temporal Growth

Plant interactions and soil community composition significantly affect native plant growth. The results suggest that the native *P. laciniata* had optimal growth in the nutrient sterilized soil while presenting its lowest growth rate in the nutrient soil (Table 1 and Figure 2).

### 3.3. Effect of Soil Diversity on the Native Plant Response

The plant growth parameters were significantly affected by the soil properties. The native target plant growth in monoculture was decreased in the nutrient soil. The results show that the native total was 93.33% higher in the nutrient sterilized soil in monoculture than in the nutrient soil. In comparison, the lowest biomass of *P. laciniata* was noted in all soil types under co-invasion. However, a better competitive ability was observed in the nutrient soil than in the nutrient sterilized soil and the natural soil (Figure 3 and Figure 4). This study shows that the negative effect of the invasive plant on the native plant biomass was more pronounced under co-invasion than under single invasion, especially in the nutrient sterilized soil (Figure 3).

### 3.4. Effects of Plants Interactions and Soil Diversity on Native Pterocypsela laciniata

Native and invasive plants evolve in the same ecosystem, competing for nutritional resources, such as light, sun, and soil nutrients. These plant–plant interactions are usually negative for native plant growth due to the strong competitive ability of invasive plants. This study shows that the presence of invasive plants in native plant communities significantly negatively affects native plant growth. The competitive interaction index (CII) shows that the negative impact of invasive plants on native plant presence depends on the soil.

This negative effect of invasive plants was more pronounced in the natural soil under single invasion than in the nutrient and sterilized soils. In contrast, the native *P. laciniata* presented its best competitive ability in the nutrient sterilized soil, being about 100% higher than in the natural and nutrient soils under single invasion and co-invasion. In addition, this study shows that the significant negative effect of co-invasion was higher in the nutrient sterilized soil than in the natural and nutrient soils. In addition, the results show that the invasive plants’ negative effects on native plant growth were more significant under *S. canadensis* invasion than under *A. subulatus* Michx invasion.

### 3.5. Effects of Plant–Soil Interaction on Native Plant Response

Interactions between plant combination and soil diversity affected the native plant growth parameter: *P. laciniata* height was negatively affected by the plant combination. In addition, the native plant produced its lowest shoot biomass in the nutrient soil and tended to have the greatest biomass in the nutrient sterilized soil. Significant differences were observed in the native plant response under *S. canadensis* and *A. subulatus* Michx. invasions (Table 1). The native plant response was specifically more significant in the nutrient soil. Significant differences in the *P. laciniata* response under the co-invasion of *S. canadensis* and *A. subulatus* Michx. were observed in the nutrient sterilized soil. Significant effects were observed on the response of the native *P. laciniata* due to the soil conditions and the planting patterns. However, no significant effect was observed on the native plant stem diameter due to the synergetic effects of the planting pattern and soil diversity (Table 2).

### 3.6. Effects of Planting Patterns on the Soil Properties

The highest pH value was noted in the natural soil in monoculture. In contrast, the highest EC was recorded in the nutrient soil under single invasion, and the highest moisture value was noted in the nutrient soil out of all soil types (Table 2). The total soil phenolic content was 12.62% higher in the natural soil under single invasion than in the CK. In comparison, the total phenolic content was higher by 10.78% under co-invasion than in the CK. No difference was noted in the values of the total phenolic content in the nutrient sterilized soil under all planting types (Table 3). The soil total flavonoid content was 28.98% higher in the natural soil than in the CK under single invasion. While a 12.34% increase in the total flavonoid content was noted in the nutrient sterilized soil compared to CK under single invasion. However, there was no difference noted in the values of the total flavonoid content in the nutrient soil under all planting types (Table 4).

## 4. Discussion

The coordination of plant functional traits with changes in the environment helps to understand the mechanisms underlying both the invasiveness and adaptation of plants [29]. Thus, to investigate the performance and functional traits of native *P. laciniata*, this experiment was conducted with a combination of soil types and invasion levels. However, plants need temperature, light, water, mineral nutrients, and support to grow to the optimum size and to produce the optimum crop [32]. The ability to adapt to the soil properties is another factor that needs more consideration. For instance, the native plant growth and competitive ability were not significant in the nutrient soil, which is characterized by a higher level of nutrients and a higher pH than the nutrient sterilized soil and the natural soil [33] (Table 1). This is because the high moisture level caused a slow growth rate in the nutrient soil compared with the natural and nutrient sterilized soils, in which the soil microbial community might be eliminated by sterilization and characterized by a normal level of moisture. In contrast, a high moisture content increases the growth of invasive species. One preceding study stated that invasive *Alternanthera philoxeroides* adapted to soil types and showed performance advantages over its native counterpart in nutrient soil compared to natural soil under well-watered conditions [29].

### 4.1. Effects of Soil Diversity on Native Plant Competitive Ability

Plants compete for limited resources in soil. Soil microorganisms, especially mycorrhizal fungi, and bacteria affect the diversity and abundance of belowground soil organisms and can thus play a significant role in the native plant response to invasion [34,35]. Previous studies have suggested that soil microbes are one of the key components that either facilitate or inhibit plant invasion [36,37,38]. This study shows that the native *P. laciniata* presents its optimal growth rate in nutrient sterilized soil, which may be characterized by the absence of soil microorganisms compared with the natural soil and the nutrient soil. This situation might be due to the absence of soil microorganisms, which could inhibit native plant growth. Moreover, the native plant height was higher under single invasion and co-invasion in the nutrient sterilized soil than in the natural and nutrient soils (Figure 2), contrary to the shoot and root biomasses, which were lower in the nutrient sterilized soil under co-invasion due to the highly competitive effects of the invasive plant, which were also in their optimal growth condition in the nutrient sterilized soil. Our results also align with findings indicate growth traits represent plant–soil interactions, contributing to the success of plant invasion. For example, positive plant–soil interactions enhance invasive plants’ growth performance, and negative plant soil interactions harm invasive species in their native range [17,39]. In addition, invasive species can also benefit from plant–soil interactions by altering biogeochemical cycling [40]. However, the association between successful invasion and plant–soil interaction raises some important questions that are still largely unanswered. In this regard, our study contributes a little by suggesting that the native plant growth under invasion depends not only on the soil condition but also on the type of invasion and the identity of the invasive plants.

### 4.2. Effects of Plant Interactions on Soil Properties

Invasive plants have the capability to change the properties of their surrounding soil. Accordingly, in turn, the changes in the soil properties can influence the performance of native and invasive species at the same time [21]. Native and invasive plants evolving in the same area interact with each other. These interactions can be competitive or synergetic, depending on the plants. In addition, it is well-known that competitive outcomes differ from one soil to another, suggesting an orientation determined by identifying context-dependent responses driven by the soil microbial communities [41]. The results show that the soil properties were affected by the plant’s interactions, especially in the nutrient and nutrient sterilized soils (Table 1). Soil pH, electrical conductivity, and moisture depend on the original soil [42], which can be later modified by a plant, causing changes in the soil’s chemical properties. High soil moisture and electrical conductivity were observed in the nutrient soil (Table 1), which might be due to the high nutrient and microbial community levels. Significant differences were observed in the soil pH, electrical conductivity, and moisture depending on the soil diversity, plant identity, and invasion type. The results show a higher soil pH and electrical conductivity in the monoculture and under single invasion with *S. canadensis* than under co-invasion with *A. subulatus* Michx. Soil pH and electrical conductivity were less significant under co-invasion than under single invasion, except for in the nutrient sterilized soil. Kaisermann et al. [43] tested whether changes in the soil conditioning phase influenced plant–plant interactions.

To address this, they compared the growth of two grass species in different planting patterns in soils with different conditioning histories. In the results, they found that competitive plant interactions were influenced by soil conditions dominated by *Leontodon hispidus*. Dong et al. [21] also noted that soil culturing conditions decreased the growth and competitive ability of both invasive *Solidao canadensis* and native species. Similarly, our results show that soil properties are more influenced by single invasion or co-invasion than monoculture. Consequently, the total soil phenolic and total flavonoid content analyses showed that the planting pattern significantly affected the soil composition (Table 2, Table 3 and Table 4). The total flavonoid content was more important in the nutrient sterilized soil than in the nutrient soil. This situation may explain the better growth rate of the three plants in the nutrient sterilized soil, in which the total amount of phenolic compounds was significantly reduced with sterilization. This is because these compounds influence plant–plant interactions, such as the germination of plants, including early root growth [44,45]. There is a relationship between changes in soil properties induced by invasive species and their feedback to native species. For instance, an increasing number of studies indicate that, after introduction, invasive plants may alter the structure and abundance of soil microbes, inhibit or stimulate soil microbial activity, and increase or decrease the rate of soil nitrogen and carbon cycling [46,47], which can have either positive or negative effects on soil biota. Some of these changes in the soil may benefit the invasive species by promoting further invasion and out-competing native plants [46]. Therefore, in our study, the nutrient sterilized soil gave positive feedback and provided better growth conditions for the studied native plants.

## 5. Conclusions

Plant–soil and plant–plant interactions are the main mechanisms underlying ecosystem diversity and structure. The nutrient sterilized soil presented the optimal growth condition for all plants, followed by the natural soil. The native plant response was more impacted by soil conditions than by the identity of the invasive plant and the type of invasion. This study shows that changes in soil properties may facilitate the success of invasive species over native species and that invasive species may change soil functioning and plant community composition via the modification of plant–soil feedback. Although outside of the scope of this study, more considerations of the soil type and quality need to be taken into account for a better understanding of plant–soil interactions and the native plant response under different degrees of invasion.

## Figures and Tables

**Figure 1 life-12-01898-f001:**
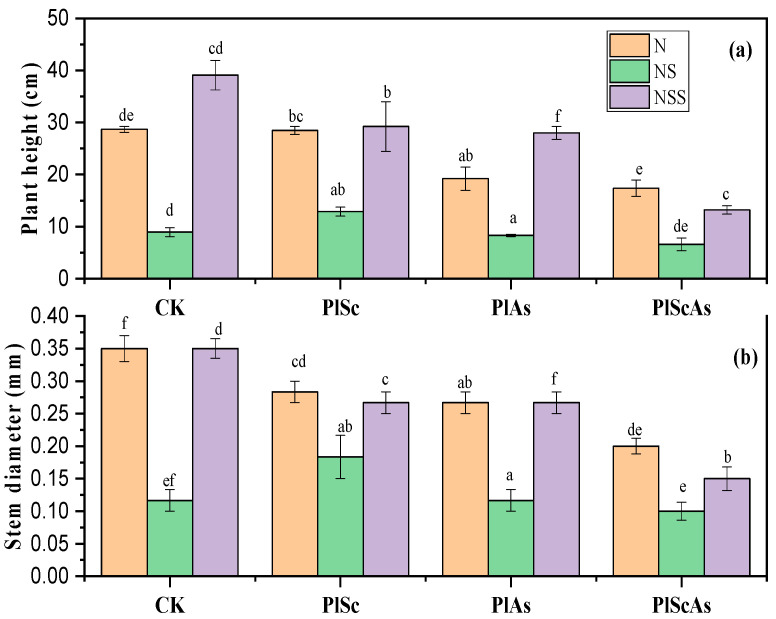
Effect of invasive plants and soil diversity on the native *P. laciniata* (**a**) plant height and (**b**) stem diameter under single invasion and co-invasion in different soils. Mean + SE with different letters indicates a significant difference among mono- and mixed-culture treatments (at *p* < 0.05). CK: *Pterocypsela laciniata* (Pl) growth in monoculture, PlSc: Pl grown under single invasion with *Solidago Canadensis*, PlAs: Pl grown under single invasion with *Aster subulatus* Michx, PlScAs: Pl grown under co-invasion of Solidago Canadensis and *Aster subulatus* Michx, N: natural soil, NS: nutrient soil, NSS: nutrient sterilized soil.

**Figure 2 life-12-01898-f002:**
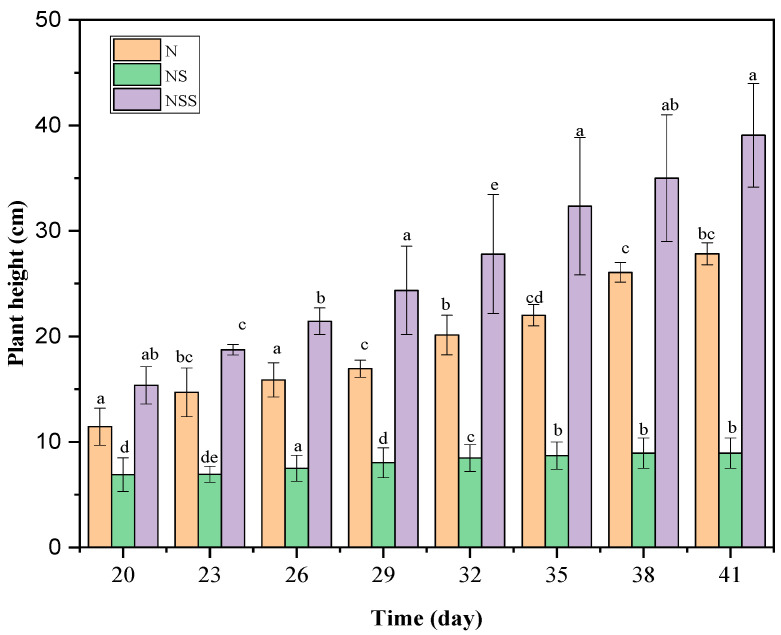
Temporal growth of the native plant *P. laciniata* under different soil conditions. Mean + SE with different letters indicates a significant difference among mono- and mixed-culture treatments (at *p* < 0.05).

**Figure 3 life-12-01898-f003:**
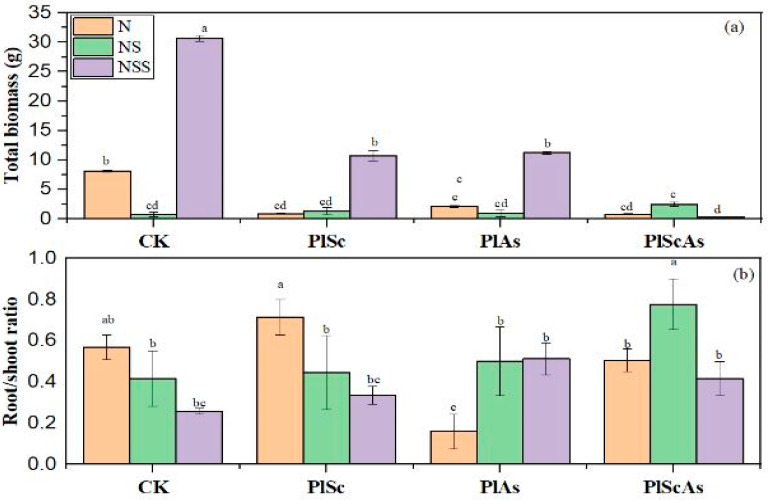
Effect of invasive plants and soil diversity on the native *P. laciniata* total biomass (**a**) and root/shoot ratio (**b**) under single invasion and co-invasion in manipulated soil conditions. Mean + SE with different letters indicates a significant difference among mono- and mixed-culture treatments (at *p* < 0.05). CK: *Pterocypsela laciniata* (Pl) growth in monoculture, PlSc: Pl grown under single invasion with *Solidago Canadensis*, PlAs: Pl grown under single invasion with *Aster subulatus* Michx, *PlScAs*: Pl grown under co-invasion of *Solidago Canadensis* and *Aster subulatus* Michx.

**Figure 4 life-12-01898-f004:**
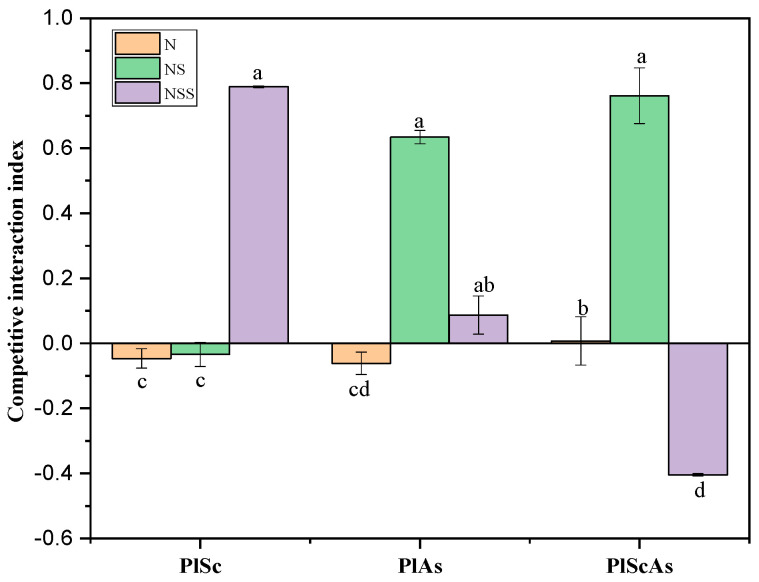
Effects of invasive identity and diversity on the native plant competitive interaction index. Mean + SE with different letters indicates a significant difference among mono- and mixed-culture treatments (at *p* < 0.05). PlSc: Pl grown under single invasion with *Solidago Canadensis*, PlAs: Pl grown under single invasion with *Aster subulatus* Michx, PlScAs: Pl grown under co-invasion of Solidago Canadensis and *Aster subulatus* Michx.

**Table 1 life-12-01898-t001:** Effects of soil conditions and invasive plant identity on the *P. laciniata* response under single invasion and co-invasion. Main ANOVA effects include plant height, stem diameter, shoot biomass, and root biomass. The *p* values are statistically significant at *p* < 0.05. The *F*-value is the ratio (*F*-ratio) of the between and within variation, and *P* is the significant level.

Factors	Plant Height		Stem Diameter		Shoot Biomass		Root Biomass	
	(cm)		(cm^2^)		(g)		(g)	
	* **F** *	* **P** *	* **F** *	* **P** *	* **F** *	* **P** *	* **F** *	* **P** *
Soil	45.8908	0.001	41.444	0.001	21.919	0.001	33.299	0.001
Species	23.775	0.001	25.444	0.001	6.076	0.010	11.700	0.001
Soil * Species	4.096	0.01	2.444	0.084	7.685	0.001	17.098	0.001

**Table 2 life-12-01898-t002:** Effects of invasive plants on soil properties. Values are means ± standard errors. Within the different soils and plants, values followed by different letters are significantly different at *p* < 0.05.

	pH			EC (µc/cm)			Moisture (%)		
Soils	N	NS	NSS	N	NS	NSS	N	NS	NSS
Planting Types									
CK	6.5 ± 0.000	5.8 ± 0.667	5.7 ± 0.44	0.033 ± 0.03	0.175 ± 0.04	0.125 ± 0.02	38.1 ± 0.38	98.7 ± 0.33	72.5 ± 0.07
PlSc	6.5 ± 0.289	5.8 ± 0.441	4.7 ± 0.44	0.035 ± 0.09	0.181 ± 0.01	0.100 ± 0.02	48.2 ± 0.23	100	75.6 ± 0.03
PlAs	5.2 ± 0.333	6.5 ± 0.000	4.7 ± 0.33	0.031 ± 0.03	0.334 ± 0.02	0.155 ± 0.02	33.7 ± 0.42	100	81.3 ± 0.02
PlScAs	4.8 ± 0.167	4.8 ± 0.441	6.0 ± 0.50	0.027 ± 0.03	0.128 ± 0.01	0.211 ± 0.05	25.2 ± 0.96	100	78.2 ± 0.05

N represents natural soil, NS represents nutrient soil, and NSS represents nutrient sterilized soil.

**Table 3 life-12-01898-t003:** Total soil phenolic under different planting systems. Values are means ± standard errors. Within the different soils and plants, values followed by different letters are significantly different at *p* < 0.05.

Soils	Natural Soil	Nutrient Soil	Nutrient Sterilized Soil
Planting Types			
CK	0.090 ± 0.009 b	0.091 ± 0.0007 b	0.057 ± 0.0035 c
PlSc	0.103 ± 0.005 ab	0.071 ± 0.005 bc	0.059 ± 0.0025 c
PlAs	0.088 ± 0.003 b	0.084 ± 0.0015 b	0.059 ± 0.00176 c
PlScAs	0.091 ± 0.007 b	0.102 ± 0.009 ab	0.059 ± 0.002 c

CK: *Pterocypsela laciniata* (Pl) growth in monoculture, PlSc: Pl grown under single invasion with *Solidago Canadensis*, PlAs: Pl grown under single invasion with *Aster subulatus* Michx, PlScAs: Pl grown under co-invasion of *Solidago Canadensis* and *Aster subulatus* Michx.

**Table 4 life-12-01898-t004:** Soil total flavonoid under different planting systems. Values are means ± standard errors. Within the different soils and plants, values followed by different letters are significantly different at *p* < 0.05.

Soils	Natural Soil	Nutrient Soil	Nutrient Sterilized Soil
Planting Types			
CK	0.049 ± 0.00274 cd	0.068 ± 0.00928 bc	0.070 ± 0.00636 b
PlSc	0.045 ± 0.00208 d	0.062 ± 0.00981 bc	0.081 ± 00.0012 ab
PlAs	0.046 ± 0.00088 d	0.068 ± 0.01271 bc	0.078 ± 0.00586 b
PlScAs	0.069 ± 0.00819 bc	0.057 ± 0.00145 c	0.073 ± 0.00817 b

CK: *Pterocypsela laciniata* (Pl) growth in monoculture, PlSc: Pl grown under single invasion with *Solidago Canadensis*, PlAs: Pl grown under single invasion with *Aster subulatus* Michx, PlScAs: Pl grown under co-invasion of *Solidago Canadensis* and *Aster subulatus* Michx.

## Data Availability

Not applicable.

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
