# Peer review of "Effect of Soil Type on Native Pterocypsela laciniata Performance under Single Invasion and Co-Invasion"

_life, 2022, doi:10.3390/life12111898_

Round 1

Reviewer 1 Report

Kindly see attached file

Author Response

Thank you very much for your valuable suggestions. We incorporated your suggestions throughout the manuscript. Please see attachment.

Thanks

Reviewer 2 Report

The authors tried to assess the response of native plants affected by single and co-invasion in different soil conditions. Although the paper has enough introduction and well-explained methods, there are still some points, especially about the discussion. 

-Bring some values in the abstract, it’s empty of percentages. Some numbers value, like how much changed under treatments by %.

-Think about another title. The current title is NOT attractive. It's like a piece of the result!

-Replace lines 150 and 151 with each other.

-Please report your results with some numbers! For example: in lines 174-179: “A better growth rate and competitive ability were observed in the nutrient sterilized soil, followed by the natural soil. In addition, more pronounced negative effects of invasive plants on the native P. laciniata were observed under A subulatus Michx invasion compared with S. canadensis. This negative effect of invasive plant presence was more significant under co-invasion than single invasion”.

Ok, How much is this significant difference? Report it by %. 

(Use this way for reporting your other results)

-Please remove Tables 2,3,4 from the discussion section and place them into the results section.

-Dissection is poorly written. It needs more extant based on your findings and comparison with other studies. It needs to use more references in this section. 

Author Response

Thank you very much for the comments. Kindly see the attachment as a response to the comments.

Thanks

Reviewer 3 Report

The manuscript focuses on the assessment of the effects of soil diversity, invasive plant identity and the type of invasion on the native plant response. The manuscript is generally well-written, however there are some sections require improvement to further enhance the quality of the manuscript. The comments are as follows: 

Abstract: 

The abstract provides sufficient essential information. 

Introduction: 

Lines 77-78: Suggest the authors to provide some reviews on the previous studies of invasive plant management control. This will further highlight the problem statement clearer and enable the readers to understand that many studies had been carried out on the aspect of invasive plants, hence neglecting the native plants. 

Lines 83-85: Suggest rewriting the objectives to make it clearer. State the objectives clearly similar to that in the Abstract section. 

Materials and Methods: 

Lines 106-115: Suggest the authors to improve the description of pot experiment setup. The authors should explain or elaborate in detailed the following items:

(a) how much soil was put in the pot?

(b) how did you sow the seeds, how many treatments (suggest displaying the list of treatments in tabular form)?

(c) how did you arrange the pots according to different treatments (no experimental design was mentioned in the manuscript)?

(d) is there any watering being done?

(e) how long did you observe the pot experiment, and at which stage you decided to stop the pot experiment, etc.? 

Results:

Figure 1: Please define the abbreviations of CK, PlSc, PlAs, and PlScAs. All these abbreviations should be defined either in the description of each table or define them in the list of treatments in the pot experiment Materials and Methods section. Please amend the same thing for other tables and figures. 

Discussion: 

The discussion section is well-discussed. 

Conclusion: 

The findings were well-concluded and suggestions for future studies were mentioned. 

Author Response

Thank you so much for your valuable comments and suggestions. Kindly see attachment file for the response to the comments.

Thanks

Round 2

Reviewer 2 Report

The authors well addressed the points, and it's suitable to be published.

Best regards.

Author Response

Thank you very much for giving your precious time to review this manuscript and give us very useful suggestions to improve our manuscript. We also thanks you to consider our manuscript for publication. 

Regards